# A Novel L-Shaped Metalens for Ultra-Wide Band (UWB) Antenna Gain Improvement

Vincenza Portosi [ID], Antonella Maria Loconsole [ID], Antonio Campana, Francesco Anelli [ID] and Francesco Prudenzano *[ID]

Department of Electrical and Information Engineering, Politecnico di Bari, Via Orabona, 4, 70125 Bari, Italy;
vincenza.portosi@poliba.it (V.P.); antonellamaria.loconsole@poliba.it (A.M.L.);
a.campana2@studenti.poliba.it (A.C.); francesco.anelli@poliba.it (F.A.)
* Correspondence: francesco.prudenzano@poliba.it

**Abstract:** In this work, a novel metamaterial lens (metalens) is designed and optimized to improve the radiation performance of an antipodal Vivaldi antenna for wideband applications. The metalens is integrated into the antenna substrate, and placed close to the tapered slot in the end-fire direction, allowing the preservation of the lightweight and compactness of the antenna. The prototype has been fabricated and characterized, demonstrating good agreement with the simulations. The insertion of the metalens allows, with respect to the pristine Vivaldi, a measured maximum gain of $G_{max} = 14.2$ dB, increased by about $\Delta G_{max} = 4.8$ dB; an operating bandwidth of $f = 3 \div 14.7$ GHz, increased by $\Delta f = 1.2$ GHz; and a radiation pattern with a maximum reduction in half-power beamwidth of $\Delta HPBW_{max} = 31.3°$, more symmetrical in the E and H planes.

**Keywords:** metamaterials; microstrip antennas; ultra-wide band antennas; Vivaldi antennas

## 1. Introduction

Ultra-wide band antennas in planar technology have attracted research interest during the last decade thanks to their high gain, wide bandwidth, compact size, high efficiency, stable radiation pattern, and low sidelobe level. In particular, planar Vivaldi antennas find application in different fields, such as ground-penetrating radar (GPR) [1], fifth-generation (5G) and ultra-wide band (UWB) communication systems [2–6], microwave imaging [7,8], and medical applications [8]. Various technologies have been proposed in the literature to fabricate low-cost and highly compact Vivaldi antennas, employing laser etching/writing [2], standard printed circuit board (PCB) technology [3–8], 3D additive manufacturing processes [9], and substrate-integrated waveguide (SIW) technology [10].

Metamaterials find applications in many fields. Lenses based on metamaterial technology (metalenses) are designed, optimized, and fabricated in order to improve antenna radiation performance [11–27]. Non-resonant metamaterials (NRMs), based on conventional parallel-line unit cells, are used to enhance the gain [15–17] and to stabilize the radiation pattern [17]. In [18], a zero-index metamaterial (ZIM) by using meander line cells is proposed for obtaining a gain increase of about 4 dB of an antipodal Vivaldi antenna (AVA). A modified H shape cell is used for a ZIM lens to obtain a gain enhancement of up to 2.6 dB of an AVA slot antenna in frequency bands close to 60 GHz [19]. Epsilon-near-zero metamaterials (ENZMs) can be exploited to increase the gain of conventional [20] and double-slot AVAs [21]. Gradient refractive index (GRI) metasurface lenses, based on non-resonant unit cells, allow the improvement in the radiation properties if placed in front of the antenna at an optimized distance [22,23].

In this work, a novel metalens is designed to improve the radiation performance of an antipodal Vivaldi antenna, operating in the $f_{AVA} = 3 \div 13.5$ GHz band. The preliminary design of the metalens has been performed with the S-parameter retrieval method (SPRM) [28,29], and the optimization of the antenna with the metalens has been performed with full-wave numerical simulations. The L-shaped geometry of the unit cell is novel. Its

geometry allows a design with a high degree of freedom with the aim of increasing the gain and maintaining, or slightly increasing, the bandwidth of an AVA. A prototype has been fabricated with the PCB process. The characterization of the fabricated prototype shows good agreement with the simulation. The measured maximum gain is increased from $G_{max} = 9.4$ dB for the case without the metalens to $G_{max} = 14.2$ dB with the metalens. Moreover, an increase of about 10% in the operation bandwidth and a maximum reduction of about 50% at $f = 14$ GHz in the half-power beamwidth is obtained.

## 2. Method and Theory

The metalens is designed to improve a pristine AVA, consisting of two symmetrical, exponentially tapered patch flares printed on the opposite sides of the dielectric substrate. The equations reported in [5] have been used for the design of the exponentially tapered flares.

Electromagnetic (EM) metamaterials are designed as periodic repetitions of elementary units with sizes significantly less than the wavelength of the propagating EM field. Therefore, the metalenses are inhomogeneous structures that can be globally considered as a homogeneous medium. Their EM properties can be described by the effective parameters, calculated considering the average of the local charge, current, and field distribution. Considering the state of the art, there are several approaches for modeling metamaterials [28–30]. The most largely employed is based on the evaluation of the volumetric effective electric permittivity and volumetric magnetic permeability [1,2,4,8,12–23]. This method requires the definition of an effective thickness $d_{eff}$ related to volumetric effective parameters of the effective homogeneous layer modeling the metamaterial [28–30]. The metalens modeling has been performed by means of the SPRM approach, based on the assumption of layer homogenization. In this work: (i) the S-parameters have been numerically simulated with the commercial EM simulation software, CST Studio Suite®; and (ii) the complex impedance $Z_{eff}$, the complex refractive index $N_{eff}$, the effective electric permittivity $\varepsilon_{eff}$, and the effective magnetic permeability $\mu_{eff}$ have been calculated from the simulated S-parameter, using the inversion algorithm based on the Kramers–Kronig relationship [28,29]. The retrieved effective parameters have been used for the preliminary design. However, the coupling effects between the metalens and the antenna cannot be neglected. Therefore, the metalens integrated into the antenna substrate has been considered in the full-wave simulations performed for the optimization [29].

## 3. Design and Results

### 3.1. Vivaldi Antenna Design

The AVA is schematically illustrated in Figure 1. It is an exponentially tapered three-flare patch antenna operating in the frequency range $f_{AVA} = 3 \div 13.5$ GHz. The shapes of the flare edges allow an enhancement in compactness [3].

In the design, the antenna optimization has been performed by numerical simulations by varying the following parameters: (i) the position of the second point of the first flare edge $d_1$; (ii) the distance between the first and the second flares $d_2$; (iii) the second flare width $d_3$; (iv) the distance between the second and the third flares $d_4$; (v) the second flare width $d_5$; and (vi) the position of the second point of the third flare edge $d_6$. The resulting geometrical parameters for the optimized antenna, allowing the gain maximization to parity of bandwidth, are listed in Table 1.

The Rogers RO4350B laminate, with a dielectric permittivity of $\varepsilon_r = 3.66$ and a loss tangent of $tan\delta = 0.0037$, having a commercial thickness of $t = 0.762$ mm and standard copper cladding with a thickness of $t_c = 0.035$ mm, has been chosen.

The simulated frequency $-10$ dB bandwidth is about $BW = 10.5$ GHz, over the frequency range $f_{AVA} = 3 \div 13.5$ GHz, and the gain is close to $G = 10$ dB for frequencies higher than $f > 7.5$ GHz.

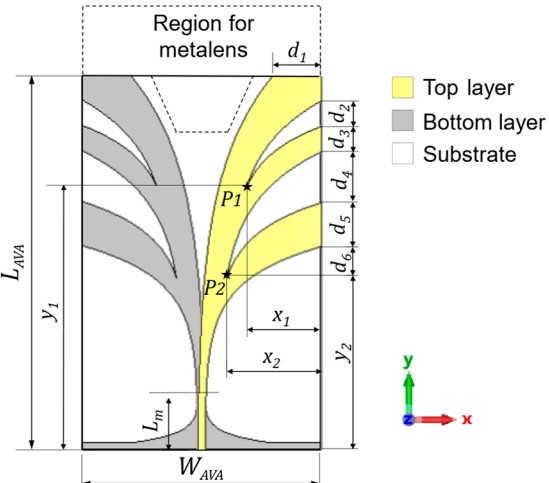

**Figure 1.** Schematic of AVA with geometrical parameters and the region (dashed contour) hosting the metalens.

**Table 1.** Geometrical parameters of the AVA.

| Parameter | Description | Value (mm) |
|:---:|:---:|:---:|
| $L_{AVA}$ | Antenna length | 73.7 |
| $W_{AVA}$ | Antenna width | 48 |
| $L_m$ | Feed microstrip length | 8.57 |
| $w_m$ | Feed microstrip width | 1.5 |
| $x_1$ | x-position of point P1 | 15 |
| $y_1$ | y-position of point P1 | 51.7 |
| $x_2$ | x-position of point P2 | 19 |
| $y_2$ | y-position of point P2 | 33.7 |
| $d_1$ | Position of the second point of the first flare edge | 9.7 |
| $d_2$ | Distance between the first and the second flares | 5 |
| $d_3$ | Second flare width | 5 |
| $d_4$ | Distance between the second and the third flares | 10 |
| $d_5$ | Second flare width | 8.7 |
| $d_6$ | Position of the second point of the third flare edge | 6.3 |
| $t$ | Substrate thickness | 0.762 |
| $t_c$ | Metal strip thickness | 0.035 |

### 3.2. Metalens Design

In order to improve the radiation characteristics of the AVA, in terms of gain and directivity, without significantly affecting the bandwidth, a novel metalens is designed.

For the preliminary design, in the first step, an equivalent homogeneous layer (EHL) has been considered in place of the metalens, with the aim of finding the optimized values of the effective parameters $\varepsilon_{eff}$ and $\mu_{eff}$ to increase the gain and maintain the bandwidth. As a second step, the design of the metalens unit cell is performed in order to obtain values of the effective parameters $\varepsilon_{eff}$ and $\mu_{eff}$ as close as possible to the optimized ones. This condition is verified via SPRM [28,29]. Finally, in the third step, the full simulation of the antenna and metamaterial is performed for the actual design refinement before the antenna fabrication. The EHL is shown in Figure 2 as a green layer.

Figure 3 shows the full-wave simulation of the gain $G$ as a function of the frequency of the AVA (solid curve) and the AVA with EHL having a thickness of $d_{EHL} = 0.832$ mm for the best values of effective permittivity $\varepsilon_{eff}$ and effective permeability $\mu_{eff}$ (dashed curve). The thickness of $d_{EHL} = 0.832$ mm corresponds to the sum of the substrate and the double metal strips thicknesses, $d_{EHL} = t + 2 \times t_c$. A metalens having an effective permittivity of

$\varepsilon_{effMG} = 5.75$ and an effective permeability of $\mu_{effMG} = 1.5$ could allow a maximum gain (MG) improvement of about $\Delta G = 4$ dB.

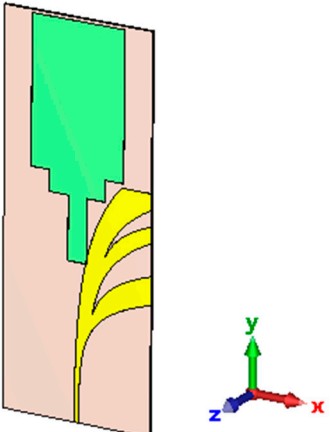

**Figure 2.** Schematic of AVA with EHL (green region) in place of the metalens.

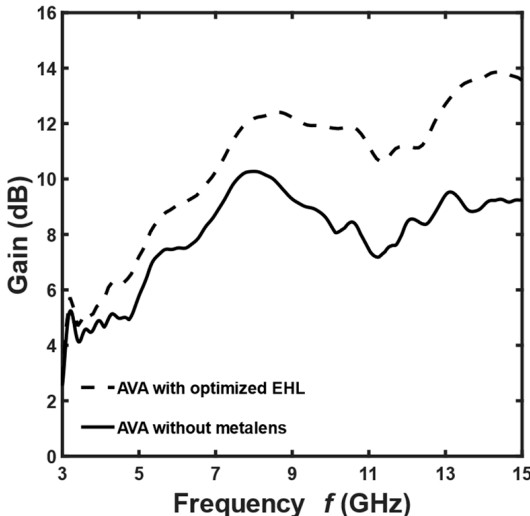

**Figure 3.** Full-wave simulation of the gain *G* as a function of frequency for AVA (solid curve) and AVA with EHL having a thickness of $d_{EHL} = 0.832$ mm for the optimized effective permittivity $\varepsilon_{effMG}$ and effective permeability $\mu_{effMG}$ (dashed curve).

A metalens based on the novel L-shaped unit cell has been optimized. Figure 4a shows the investigated unit cell. The geometry consists of an arrangement of L-shaped metal elements printed on both sides of the antenna substrate. In the proposed unit cell structure, it is possible to recognize an external square ring with four splits and two internal stubs. Different inductive and capacitive effects are induced depending on the electric field orientation of the traveling wave with respect to the metalens plane.

Considering the electric field (E-field) polarized along the *x*-axis, inductive effects along the metal strips and capacitive effects at the gaps occur, as shown in Figure 4a. The equivalent inductors, namely *L*, represent the self-inductance produced by the metal L-shaped elements. The equivalent capacitances originate from the electric charges accumulated via the splits, namely $C_1$, and by the coupled charges between the adjacent metal L-shaped elements, namely $C_2$, $C_3$, and $C_4$ [31]. The equivalent circuit of the L-shaped unit cell is depicted in Figure 4b; its effective complex impedance $Z_{eff}$ can be fine-tuned by varying the geometries of the conductive metal inclusions, the gap width between them, and the dielectric permittivity of the substrate [32,33].

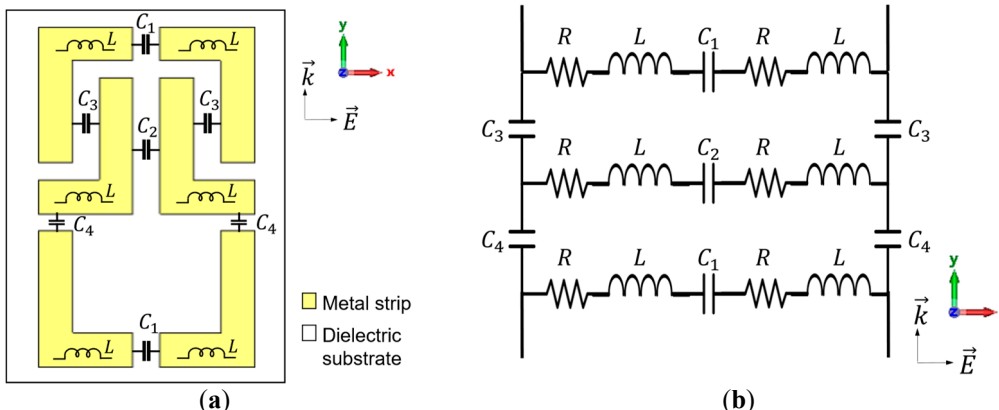

**Figure 4.** (**a**) Schematic of the designed L-shaped unit cell and (**b**) its equivalent circuit.

For the SPRM approach, the S-parameters simulations of the stand-alone unit cell, i.e., before its integration with AVA geometry, are performed by considering the same substrate of the antenna. The top and the 3D views of the L-shaped unit cell with the relative geometric parameters are shown in Figure 5a,b, respectively.

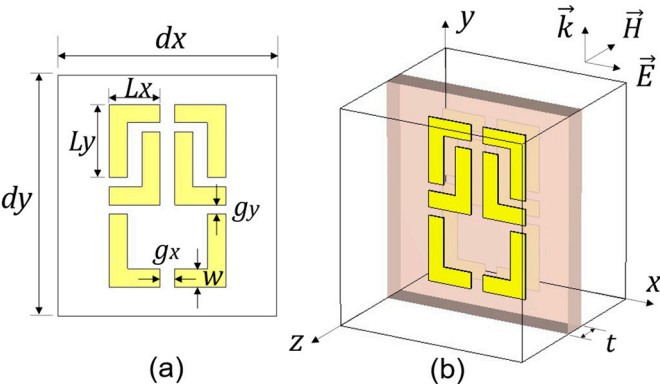

**Figure 5.** Schematic of L-shaped unit cell: (**a**) top view, and (**b**) 3D view.

In the second design step, in order to retrieve the effective parameters of the metalens, the unit cell illustrated in Figure 6 has been considered. In the simulation, the $\vec{E}\,\vec{k}$-plane of the unit cell corresponds to the Vivaldi *xy*-plane; see Figures 1 and 6. The plane EM wave excitation is obtained (see Figure 6) via the Waveguide Port 1, with open boundary conditions applied in the propagation direction. The perfect electric conductor (PEC) and the perfect magnetic conductor (PMC) boundary conditions are suitably applied.

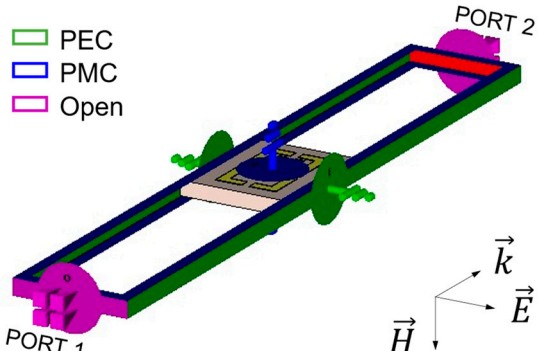

**Figure 6.** Schematic of the simulation model for the second design step of the L-shaped unit cell.

The L-shaped unit cell geometry has been optimized in order to obtain a broadband gain enhancement to parity of the bandwidth [15,16]. The geometry of the proposed unit cell offers a high degree of freedom for metalens optimization. The inductive effects can be tuned in different ways, depending on the EM polarization, by varying the length of the L-element independently along the x-direction, $L_x$, and y-direction, $L_y$. Similarly, the capacitive effects can be tuned by varying the gap widths of the splits $g_x$ and $g_y$. Moreover, changing the metal strip width $w$ affects both the capacitive and inductive effects. Therefore, the L-shaped unit cell resonant behavior has been evaluated by changing all the above-mentioned geometrical parameters and the lattice lengths $d_x$ and $d_y$.

For shortness, Table 2 directly reports the optimized values obtained in the third design step of the unit cell of the metalens integrated into the antenna. To identify these parameters, the exact number and distribution of cells, described in Section 3.3, have been considered.

**Table 2.** Geometrical parameters of the L-shaped unit cell.

| Parameter | Description | Value (mm) |
|---|---|---|
| $L_x$ | L-element length along x-axis | 1.40 |
| $L_y$ | L-element length along y-axis | 2.00 |
| $w$ | Metal strip width | 0.50 |
| $g_x$ | Distance between two adjacent metal elements along x-axis | 0.40 |
| $g_y$ | Distance between two adjacent metal elements along y-axis | 0.25 |
| $dx$ | Lattice length along x-axis | 6.00 |
| $dy$ | Lattice length along y-axis | 6.60 |
| $t$ | Substrate thickness | 0.762 |
| $t_c$ | Metal strip thickness | 0.035 |

To give an insight into the metalens behavior, the simulated amplitudes Figure 7a and phases Figure 7b of the $S_{11}$ and $S_{21}$ parameters of the L-shaped unit cell versus the frequency are shown in Figure 7 for the optimized parameters of Table 2.

The average values of the real part of the effective electric permittivity $Re(\varepsilon_{eff,av}) = 4.98$ and of the effective magnetic permeability $Re(\mu_{eff,av}) = 0.75$, retrieved with SPRM from the aforesaid $S_{11}$ and $S_{21}$ curves, are close enough to the theoretical optimized values $\varepsilon_{effMG} = 5.75$ and $\mu_{effMG} = 1.5$ identified in the first design step and provide an almost constant value of the effective refractive index over the whole frequency range. It is reported in Figure 7c. A congruent number of unit cells allows better results, taking into account the couplings between adjacent unit cells [34]. Therefore, a single cell, $3 \times 3$, and $6 \times 6$ clusters of unit cells have been also investigated, obtaining practically the same results. As a consequence, the frequency range $f = 3 \div 15$ GHz can be considered a potential operation region for the antenna. This means that the L-shaped metalens can improve the antenna radiation performance in the whole band [15,16].

Indeed, the designed lens has a higher average refractive index than that of the antenna substrate, so a waveguiding effect is possible. The EM field is better confined via the lens, enhancing the radiation beam in the end-fire direction. In addition, the high refractive index region obtained with the metalens can be used to change the EM wavefronts from spherical to approximately planar ones, resulting in a more confined radiation pattern and higher directivity. This is explained by considering that the phase velocity of the wavefront in the central region is smaller than that of the wavefront edges due to the higher refractive index metalens.

The approximated equivalent circuit reported in Figure 4b models the proposed unit cell behavior. By exploiting the Advanced Design System (ADS) [34] and making use of the tuning module, the following set of values, $L = 0.22$ nH, $C_1 = 1.6$ pF, $C_2 = 0.3$ pF, $C_3 = 1.2$ pF, $C_4 = 0.55$ pF, and $R = 2.625$ $\Omega$, have been identified, leading to a modulus of the scattering parameter $|S_{11}|$ in good agreement with that of Figure 7a.

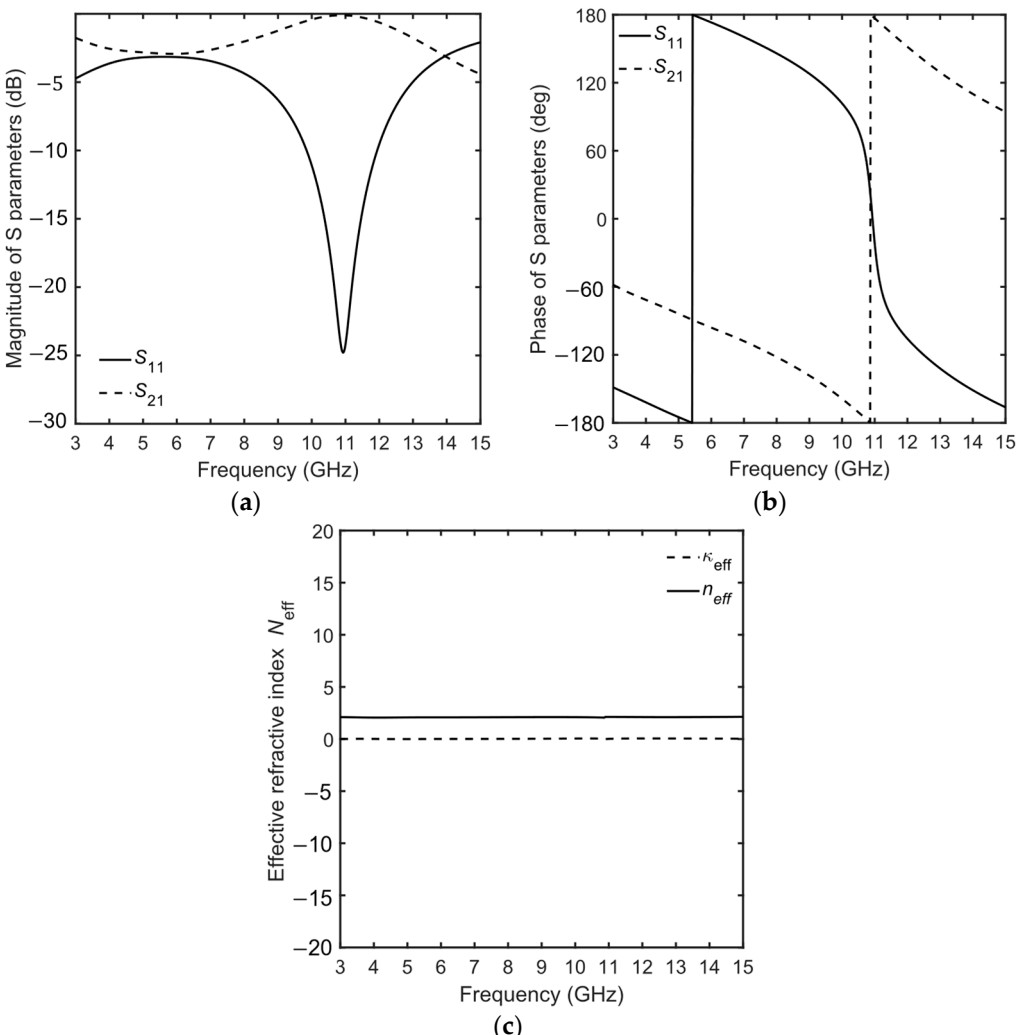

**Figure 7.** (**a**) Magnitude and (**b**) phase of the simulated S-parameters and retrieved effective parameters of L-shaped unit cell as a function of frequency. (**c**) Refractive index $N_{eff}$.

### 3.3. AVA with Metalens Design

In the third design step, the metalens has been integrated into the substrate of the antenna, and placed close to the tapered slot in the end-fire direction, as shown in Figure 1, for the optimization by numerical full-wave simulations in the frequency band $f = 3 \div 15$ GHz, in order to refine the results, by improving the realized gain of the antenna without reducing its bandwidth.

The optimized values reported in Table 2 have been found via full-wave simulations with a trial-and-error approach. Then, further optimization of the antenna with metalens has been performed by varying, in the full-wave simulation, the number and distribution of the unit cells and the distance along the *y*-direction between the metalens and antenna. Several configurations have been simulated and compared.

In Figure 8a, the metalens of the optimized AVA L-shaped#1 consists of an array of 41 L-shaped unit cells integrated on the antenna substrate, increasing the antenna length along the *y*-axis by $L_y = 47$ mm. This antenna exhibited particularly promising performance over the whole band, in particular at high frequencies.

The AVA L-shaped#2, shown in Figure 8b, consists of an array of 67 L-shaped unit cells integrated into the antenna substrate; the larger metalens also requires a slight increase in the antenna substrate width along the *x*-axis. The AVA L-shaped#2 allows better performance at low frequencies.

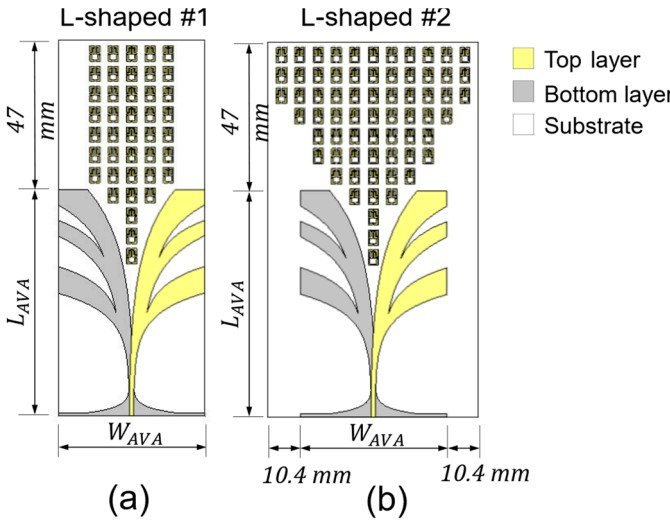

**Figure 8.** Schematic of AVA with L-shaped metalenses: (**a**) AVA L-shaped#1—array of 41 L-shaped unit cells, and (**b**) AVA L-shaped#2—triangular array of 67 L-shaped unit cells.

In Figure 9a, the modulus of the simulated scattering parameter $|S_{11}|$ as a function of frequency is illustrated for the AVA without a metalens (black curve) and the AVA L-shaped#1 (magenta curve) and AVA L-shaped#2 (green curve) metalenses. The bandwidth $f_{avaL\#1} = 3 \div 15$ GHz of AVA L-shaped#1 is increased with respect to $f_{AVA} = 3 \div 13.5$ GHz of the AVA without a metalens. The L-shaped#2 allows an improvement in impedance matching for frequencies of $f < 6.5$ GHz but reduces the $-10$ dB bandwidth to the range of $f_{avaL\#2} = 3 \div 11.8$ GHz.

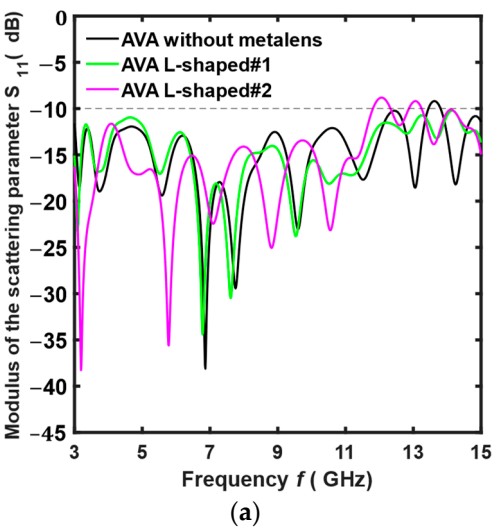

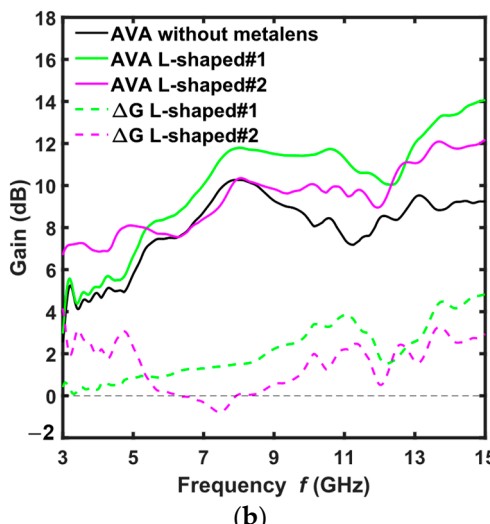

**Figure 9.** Full-wave simulation of the AVA with different L-shaped metalenses: (**a**) modulus of the scattering parameter $|S_{11}|$ as a function of frequency, and (**b**) realized gain $G$ and gain increase $\Delta G$ as a function of frequency.

In Figure 9b, the realized gain $G$ (solid curves) of the three antennas and the gain increase $\Delta G$ (dashed curves) with respect to the gain without a metalens, simulated as a function of the frequency, are reported, with the same choice for the curve colors. For the AVA L-shaped#1, the gain increase can be observed over the whole frequency range, with an average value of about $\Delta G_{av} = 2$ dB and a maximum of $\Delta G_{max} = 4.8$ dB at a frequency of $f = 15$ GHz. The maximum gain is $G_{max} = 14.1$ dB at a frequency of $f = 15$ GHz. For the AVA L-shaped#2, a gain increase of about $\Delta G = 2$ dB is obtained for frequencies of

$f < 6$ GHz and $f > 8$ GHz. The simulated maximum gain is $G_{max} = 12.2$ dB at a frequency of $f = 15$ GHz.

The L-shaped#1 metalens improves both the antenna bandwidth and the gain in the whole band, especially at higher frequencies. The AVA L-shaped#1 at higher frequencies irradiates a beam more confined than that of AVA L-shaped#2. The L-shaped#2 metalens allows a better gain improvement at the lower frequencies but reduces the bandwidth at higher frequencies.

The simulated E-field distributions in the $xy$-plane of the AVA without a metalens and AVA L-shaped#1 at a frequency of $f = 14$ GHz are shown in Figure 10. As expected, the near-field of the antenna with a metalens exhibits wavefronts that are flatter than those of the AVA without a metalens, and the radiation in the end-fire direction is increased. Moreover, the L-shaped#1 metalens reduces the antenna back propagation. The simulated 3D view of the normalized radiation patterns at a frequency of $f = 14$ GHz for the AVA and AVA L-shaped#1 are compared in Figure 11. The lens improves the antenna's directivity and reduces the half-power bandwidth (HPBW).

To conclude, the L-shaped#1 metalens is chosen as an interesting solution for prototype fabrication.

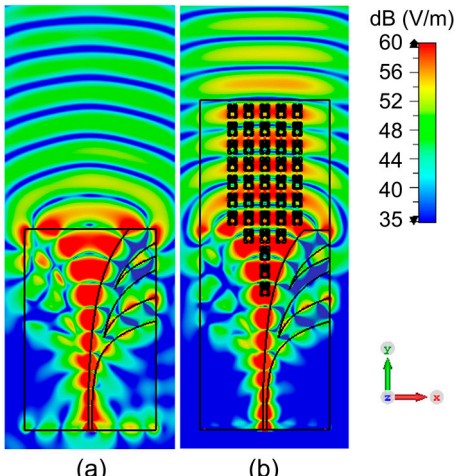

(a)      (b)

**Figure 10.** Full-wave simulation of the E-field distribution in $xy$-plane at $f = 14$ GHz of (**a**) AVA without metalens and (**b**) AVA-L-shaped#1.

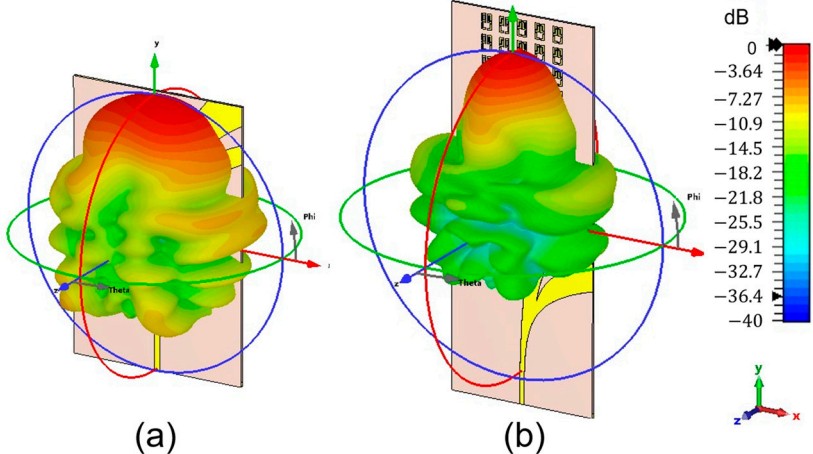

(a)      (b)

**Figure 11.** Full-wave simulation of the normalized radiation patterns 3D view at frequency $f = 14$ GHz of (**a**) AVA without metalens and (**b**) AVA L-shaped#1.

To verify the effects of the metalens on the antenna performances, the full-wave simulation of the gain $G$ as a function of the frequency for AVA L-shaped#1 (solid curve) and AVA with EHL, having a thickness of $d_{EHL} = 0.832$ mm and values of permittivity $\varepsilon_r = 4.98$ and permeability $\mu_r = 0.75$ (dashed curve) are compared in Figure 12. The dimensions in the $x$-direction and $y$-direction of the EHL are the same for the metalens L-shaped#1, and the thickness $d_{EHL}$ corresponds to the sum of the substrate and double metal strips thicknesses. A very good agreement is evident.

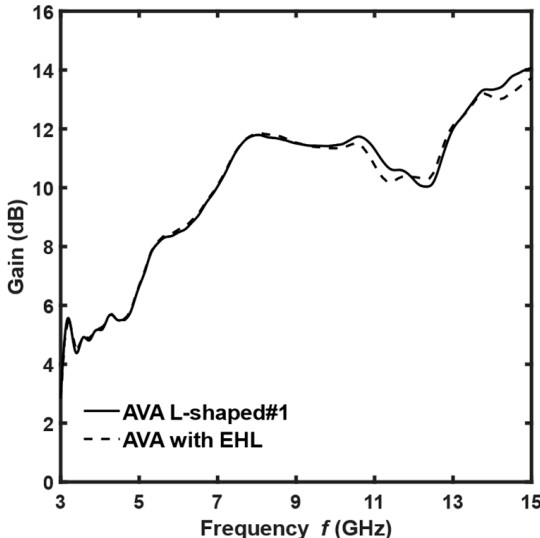

**Figure 12.** Full-wave simulation of the gain $G$ as a function of frequency for AVA L-shaped#1 (solid curve) and AVA with EHL, having thickness $d_{EHL} = 0.832$ mm and values of permittivity $\varepsilon_r = 4.98$ and permeability $\mu_r = 0.75$ (dashed curve).

## 4. Experimental Section

Antenna prototypes, AVA, and AVA L-shaped#1 have been fabricated with Rogers RO4350B dielectric substrate, with $\varepsilon_r = 3.66$ and $tan\delta = 0.0037$, by using the standard PCB process. The prototype of AVA is shown in Figure 13a. The prototype AVA L-shaped#1 is shown in Figure 13b. The scattering parameter $S_{11}$, input impedance $Z_{ant}$, and voltage standing-wave ratio (*VSWR*) versus the frequency of the two antennas have been measured with the Agilent Technologies N5224A PNA Network Analyzer. The calibration of the PNA has been performed with an N4693A 2-Port Electronic Calibration Module, setting the IF bandwidth to $IF_{BW} = 1$ KHz and the stimulus power to $P_s = -10$ dBm. The uncertainty in the measured $S_{11}$ parameter is $\pm0.04$ dB for the magnitude and $\pm0.264°$ for the phase. The radiation performance characterization has been performed with the antenna measurement system in an anechoic chamber, StarLab SATIMO. The system has been calibrated in gain with reference horn antennas. The uncertainty band of the instrument is $e_{gain} = \pm0.7$ dB. For the measurements, we have set a step of $\Delta f = 10$ MHz for linear frequency distribution and a grid size of $\Delta\theta = \Delta\phi = 4.5°$ for spatial resolution.

The simulated (solid curves) and the measured (dotted curves) moduli of the scattering parameter $S_{11}$ as a function of the frequency, for the AVA (black curves) and AVA L-shaped#1 (green curves) are illustrated in Figure 14a. The metalens allows an increase in the antenna bandwidth at high frequencies. A band of $f_{AVA} = 3 \div 13.5$ GHz for the AVA and an increased band of $f_{avaL\#1} = 3 \div 14.7$ GHz for the AVA L-shaped#1 have been measured. The simulated and measured curves are in good agreement.

The measured VSWR of AVA and AVA L-shaped#1 prototypes is $VSWR < 2$ in almost the whole operating frequency band. The introduction of the metalens reduces the VSWR for frequencies $f > 9.5$ GHz.

The measured antenna impedance of both prototypes varies around the characteristic impedance of the feed line $Z_c = 50\ \Omega$. The introduction of metalens slightly reduces the

impedance mismatch at high frequencies, in agreement with the measured parameter $S_{11}$ reported in Figure 14a.

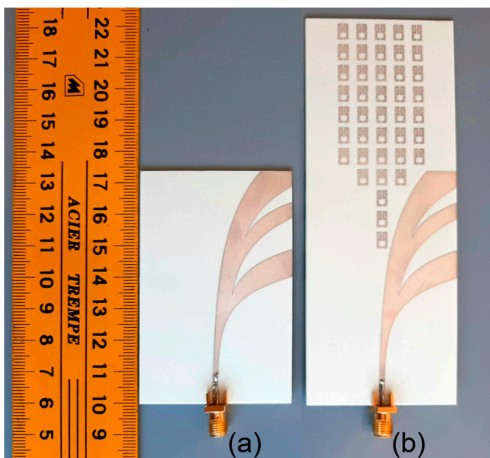

**Figure 13.** Prototypes of antipodal Vivaldi antenna without and with metalens: (**a**) AVA, and (**b**) AVA L-shaped#1.

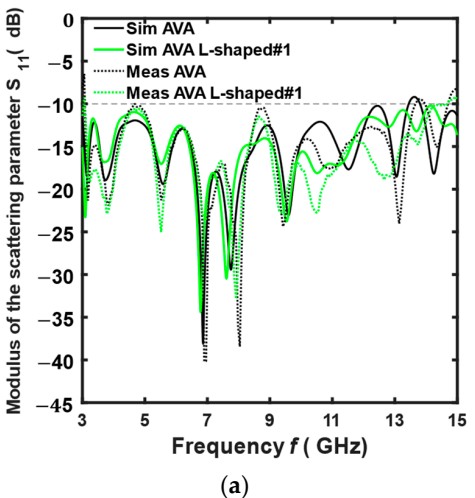

(**a**)

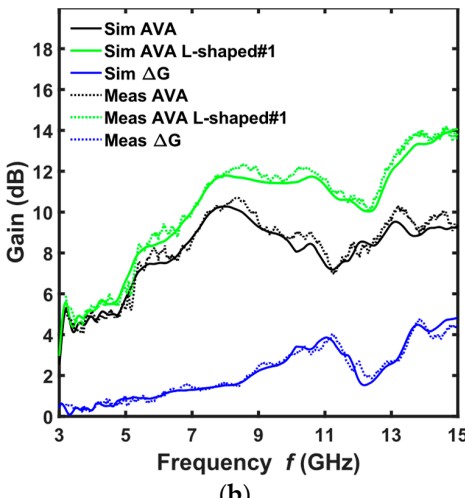

(**b**)

**Figure 14.** Comparison between full-wave simulation (solid curves) and measurement (dotted curves) of the AVA (black curves), and AVA L-shaped#1 (green curves): (**a**) modulus of the scattering parameter $|S_{11}|$ as a function of frequency, and (**b**) realized gain ($G$) and gain increase ($\Delta G$) of frequency.

The simulated and measured gain of AVA and AVA L-shaped#1 as a function of the frequency is illustrated in Figure 14b. A maximum gain $G_{max} = 10.7$ dB at a frequency of $f = 8.4$ GHz for the AVA and $G_{max} = 14.21$ dB at a frequency of $f = 14.6$ GHz for the AVA L-shaped#1 have been measured. The AVA L-shaped#1 has an improved gain with respect to the AVA over the whole bandwidth, especially for the frequencies $f > 8$ GHz, where the measured average gain increase is about $\Delta G_{avg} = 3$ dB and the maximum measured gain increase is $\Delta G_{max} = 4.8$ dB at a frequency of $f = 13.8$ GHz. Additionally, for the gain, a good agreement between the simulation and measurement is obtained.

The E-plane and H-plane radiation patterns of the antenna prototypes have been measured and compared in the whole operating frequency band. The measured E-plane (solid curve) and H-plane (dashed curve) half-power bandwidths of the AVA (black curve) and AVA L-shaped#1 (green curve) as a function of the frequency are illustrated in Figure 15. The HPBW of the antenna with metalens is reduced over the whole band in both planes, with a maximum HPBW reduction of $\Delta HPBWmax = 31.3°$ at a frequency of $f = 14$ GHz

for the E plane and of $\Delta HPBWmax = 28.3°$ at a frequency of $f = 5\,\text{GHz}$ for the H plane. The difference between the beamwidths in the E and H planes is reduced after the insertion of the metalens, hence the radiation pattern is more symmetrical. Figure 16 shows the E-plane and H-plane radiation patterns of AVA (dashed curve) and AVA L-shaped#1 (full curve), measured at frequencies $f = 14\,\text{GHz}$ and $f = 5\,\text{GHz}$ where the maximum HPBW reduction occurs.

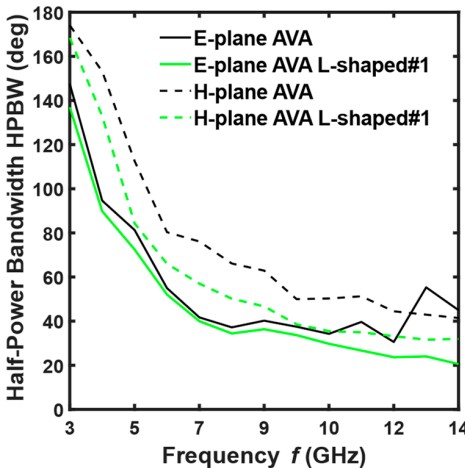

**Figure 15.** Measured E-plane (solid curve) and H-plane (dashed curve) half-power bandwidth of AVA (black curve) and AVA L-shaped#1 (green curve) as a function of frequency.

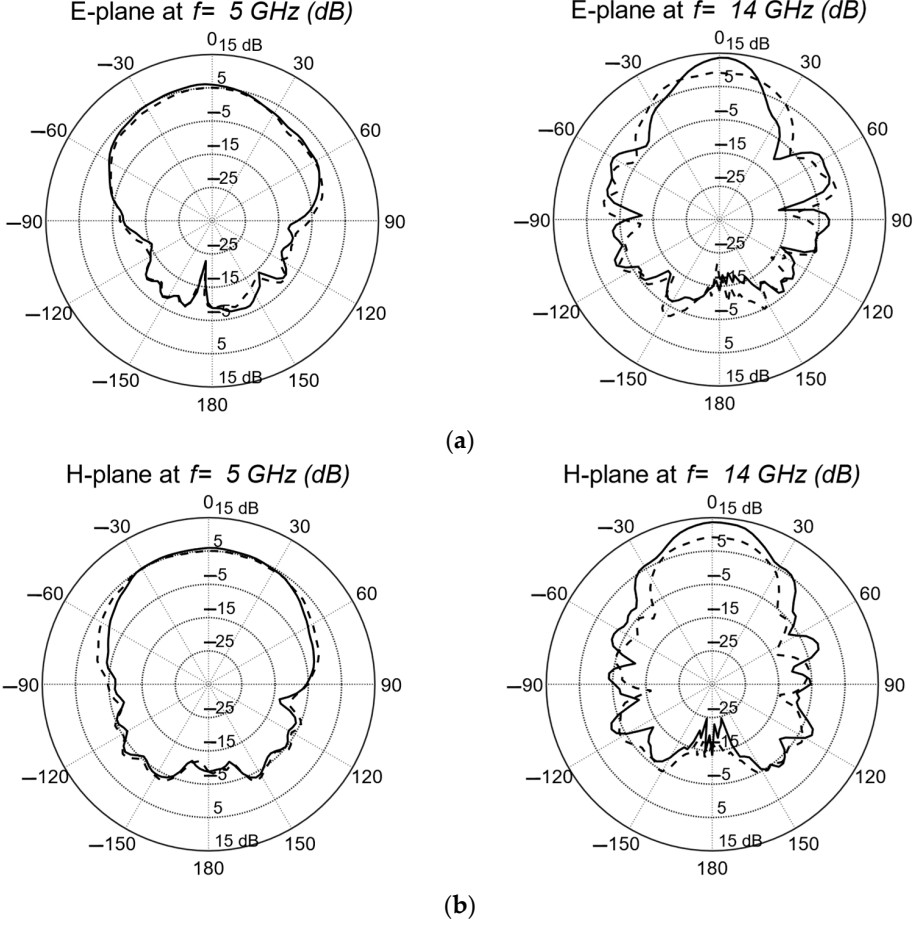

**Figure 16.** Measured (**a**) E-plane and (**b**) H-plane radiation patterns of AVA L-shaped#1 (solid curve) and AVA (dashed curve) at frequencies of $f_1 = 5\,\text{GHz}$ and $f_2 = 14\,\text{GHz}$.

A comparison between AVA L-shaped#1 and recent similar Vivaldi antennas with metalens is shown in Table 3. The comparison with the literature is not trivial if size, bandwidth, and gain are simultaneously considered. However, by comparing the measured performances of the proposed AVA L-shaped#1 with the state of the art, the size is smaller to parity of gain. As an example, the size of the AVA with a metalens [12] is $120 \times 260$ mm, and a director element is employed in addition to the metalens. In this paper, the size of AVA L-shaped#1 is $48 \times 121$ mm, without a director.

**Table 3.** Experimental results comparison of the metalens investigated in this work and similar published work.

| Ref | Size (mm×mm) ($\lambda \times \lambda$) [(1)] | Operating Bandwidth (GHz) | Min–Max Gain G (dB) | Min–Max Gain Increase $\Delta G$ (dB) | Min–Max $\Delta HPBW$ (deg) |
|---|---|---|---|---|---|
| [1] | $286 \times 300$ $1.3\lambda \times 1.4\lambda$ | $0.7 \div 2.1$ | nr [(2)] | $1.0 \div 1.9$ | E plane: $4.8° \div 9.9°$ H plane: $14.7° \div 30.3°$ |
| [2] | $200 \times 390$ $2.8\lambda \times 5.5\lambda$ | $1 \div 7.5$ | nr | $0.1 \div 4.8$ | nr |
| [4] | $138.2 \times 69.6 \times 31.7$ $7\lambda \times 3.5\lambda \times 1.6\lambda$ | $7.55 \div 22.85$ | $6.98 \div 11.54$ | nr | nr |
| [12] | $120 \times 260$ $5.8\lambda \times 12\lambda$ | $1 \div 28$ | $4.9 \div 14.6$ | nr | nr |
| [13] | $60 \times 28.6$ $5.3\lambda \times 2.5\lambda$ | $24.15 \div 28.5$ | $0.7 \div 14$ | nr | nr |
| [16] | $42 \times 78$ $1.7\lambda \times 3.1\lambda$ | $6 \div 18$ | $7.2 \div 12$ | $1.3 \div 3.6$ | E-plane: $-3° \div 17°$ H plane: $23° \div 33°$ |
| [17] | $52 \times 93$ $2.1\lambda \times 3.7\lambda$ | $6 \div 18$ | nr | $1 \div 2.5$ | nr |
| [18] | $140 \times 96$ $3.1\lambda \times 2.1\lambda$ | $1.4 \div 12$ | $9.5 \div 12.4$ | $2.5 \div 4$ | E-plane: $11.3° \div 23.2°$ H plane: $-24° \div 33.2°$ |
| [21] | $80 \times 148.5$ $1.75\lambda \times 3.2\lambda$ | $1.13 \div 12$ | $0.7 \div 14$ | nr | nr |
| **This work** | $48 \times 120$ $1.4\lambda \times 3.5\lambda$ | $3 \div 14.7$ | $4.5 \div 14.2$ | $0.2 \div 4.8$ | E-plane: $1.7° \div 31.3°$ H plane: $5.7° \div 28.3°$ |

[(1)] $\lambda$ represents the wavelength at the center frequency of the operating bandwidth. [(2)] nr: not reported.

## 5. Conclusions

Metalenses have been designed and optimized in the frequency band of $f = 3 \div 15$ GHz in order to enhance the radiation performance of an antipodal Vivaldi antenna. The metalens L-shaped#1 improves both the pristine AVA bandwidth and the gain, especially at high frequencies. A maximum gain increase of $\Delta G_{max} = 4.8$ dB and a maximum HPBW reduction of $\Delta HPBW_{max} = 31.3°$ have been measured at the frequency of $f = 13.8$ GHz with respect to the pristine AVA. A frequency band of $f_{avaL\#1} = 3 \div 14.7$ GHz, slightly larger than that of the pristine AVA, has been measured. The metalens L-shaped#2 can give better results at lower frequencies. The experimental results are in good agreement with the simulations.

**Author Contributions:** Conceptualization, investigation, and methodology, V.P., A.M.L., A.C., F.A. and F.P.; writing—original draft preparation, V.P. and F.P.; writing—review and editing, V.P., A.M.L., F.A. and F.P.; supervision, F.P. All authors have read and agreed to the published version of the manuscript.

**Funding:** This work was partially supported by the European Union under the Italian National Recovery and Resilience Plan (NRRP) of NextGenerationEU, with particular reference to the partnership on "Telecommunications of the Future" (PE00000001—program "RESTART", CUP: D93C22000910001)—STRUCTURAL PROJECT MINDS DREAMS MIllimeter wave aNtennas Disruptive Solutions and Devices foR mixing, dEtection And Manipulation of mm Waves. It was also supported by MIUR "Agriculture Green & Digital—AGREED", PNR 2015/20, n. ARS01_00254; H2020-ICT-37-2020 "Photonic Accurate and Portable Sensor Systems Exploiting Photo-Acoustic and Photo-Thermal Based Spectroscopy for Real-Time Outdoor Air Pollution Monitoring—PASSEPARTOUT" n. 101016956.

**Institutional Review Board Statement:** Not applicable.

**Informed Consent Statement:** Not applicable.

**Data Availability Statement:** Not applicable.

**Conflicts of Interest:** The authors declare no conflict of interest.

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
