# Peer review of "A Novel L-Shaped Metalens for Ultra-Wide Band (UWB) Antenna Gain Improvement"

_applsci, doi:10.3390/app13084802_

Round 1

Reviewer 1 Report

1.     Author has shown radiation pattern for 5GHz and 14 GHz, they may also study results for other resonance frequencies.

2.     Author may check in line 139-141 whether C1 are same like C2, C3, C4 capacitance or it represents capacitances due to charge accumulation and coupled charges both.

How is the proposed work different from [a], where a SRR/CSRR is explored with inherent band notch and yet yields a UWB response.

[a] “Circular SRR shaped UWB Antenna with WiMAX Band Notch Characteristics”, IEEE Radio and Antenna Days of the Indian Ocean (IEEE RADIO 2018), Mauritius, October 15-19, 2018

3.     Author may explain Fig 7, does metalens (unit cell) are working for entire bandwidth?

4.     Author may also show results for metalens array for better understanding of its performance and intercoupling between unit cells.

Author Response

Dear reviewer,

I would like to thank you for the useful and specific comments. We followed with great care all the suggestions to improve the paper. For the sake of completeness, please find attached our punctual reply to all reviewers. The text changes are underlined.

Best Regards,

Francesco Prudenzano

Reviewer 2 Report

The paper calls: "A Novel L-shaped Metalens for UltraWideBand (UWB) Antenna Gain Improvement" and concerned of new design antipodal Vivaldi antenna for wide band application (for 5G data exchange). Such result is achieved by a novel metamaterial lens. The advantage of article interesting topic of research and long references list (31 ones).

However,

The main questions to article are parameters of such emitter:

1. Impedance of such antenna?

2. Standing wave ration in working band?

Besides, the manuscript absent some data:

What about standing wave ration (SWR) of such antenna in working band?

Author Response

(The authors gave the same response as above.)

Reviewer 3 Report

This paper presents A Novel L-shaped Metalens for UltraWideBand (UWB) Antenna Gain Improvement. The novelty of this manuscript is that the metalens is integrated in the antenna substrate to improve the radiation performance of an antipodal Vivaldi antenna. The design and optimization process of structural parameters is introduced in detail. In my opinion, the work fits within the scope of Applied Sciences and should be published after minor revisions. The following are several recommendations and clarifications that must be addressed before publication.

1.      It is suggested to add a colorbar in Figure 11; The sharpness of Figures 10 and 11 is poor. It is recommended to replace them with high-definition pictures.

2.      Please add the values of capacitance, inductance, and resistance used in the equivalent circuit in Figure 4.

3.      The author needs to supplement the key parameter Settings in the Experimental section and add error analysis to make the article more complete.

4.      The presented topic is highly related to EM manipulations of metasurfaces, the following recent papers could be included in the introduction:

[1] L. Chen, Q. Ma, S. S. Luo, F. J. Ye, H. Y. Cui, and T. J. Cui, “Touch-Programmable Metasurface for Various Electromagnetic Manipulations and Encryptions,” Small, pp. e2203871, Sep 15, 2022.

[2] Q. Ma et al., "Directly wireless communication of human minds via non-invasive brain-computer-metasurface platform," eLight, vol. 2, no. 1, 2022, doi: 10.1186/s43593-022-00019-x.

[3] Li, Y., Chen, S., Liang, H. et al. Ultracompact multifunctional metalens visor for augmented reality displays. PhotoniX 3, 29 (2022). https://doi.org/10.1186/s43074-022-00075-z

Author Response

(The authors gave the same response as above.)
